# Predicted 3D model of the M protein of Porcine Epidemic Diarrhea Virus and analysis of its immunogenic potential

**Alan Rodríguez-Enríquez**[1,2☯], **Irma Herrera-Camacho**[1☯], **Lourdes Millán-Pérez-Peña**[1‡], **Julio Reyes-Leyva**[3‡], **Gerardo Santos-López**[3‡], **José Francisco Rivera-Benítez**[4☯], **Nora Hilda Rosas-Murrieta**[1☯]*

**1** Laboratory of Biochemistry and Molecular Biology, Chemistry Center, Institute of Science., Benemérita Universidad Autónoma de Puebla, Puebla, México, **2** Postgraduate in Chemical Sciences, Benemérita Universidad Autónoma de Puebla, Puebla, México, **3** Molecular Biology and Virology Laboratory, Centro de Investigación Biomédica de Oriente, Instituto Mexicano del Seguro Social (IMSS), Metepec, México, México, **4** Centro Nacional de Investigación Disciplinaria en Salud Animal e Inocuidad, Instituto Nacional de Investigaciones Forestales, Agrícolas y Pecuarias, Ciudad de México, México

☯ These authors contributed equally to this work.
‡ LMPP, JRL and GSL also contributed equally to this work.
* nora.rosas@correo.buap.mx

**Data Availability Statement:** All relevant data are within the paper and its Supporting Information files.

## Abstract

The membrane protein M of the Porcine Epidemic Diarrhea Virus (PEDV) is the most abundant component of the viral envelope. The M protein plays a central role in the morphogenesis and assembly of the virus through protein interactions of the M-M, M-Spike (S) and M-nucleocapsid (N) type. The M protein is known to induce protective antibodies in pigs and to participate in the antagonistic response of the cellular antiviral system coordinated by the type I and type III interferon pathways. The 3D structure of the PEDV M protein is still unknown. The present work exposes a predicted 3D model of the M protein generated using the Robetta protocol. The M protein model is organized into a transmembrane and a globular region. The obtained 3D model of the PEDV M protein was compared with 3D models of the SARS-CoV-2 M protein created using neural networks and with initial machine learning-based models created using trRosetta. The 3D model of the present study predicted four linear B-cell epitopes (RSVNASSGTG and KHGDYSAVSNPSALT peptides are noteworthy), six discontinuous B-cell epitopes, forty weak binding and fourteen strong binding T-cell epitopes in the CV777 M protein. A high degree of conservation of the epitopes predicted in the PEDV M protein was observed among different PEDV strains isolated in different countries. The data suggest that the M protein could be a potential candidate for the development of new treatments or strategies that activate protective cellular mechanisms against viral diseases.

**Funding:** Grant number: JFR-B and IH-C (FONSEC SADER-CONACYT 2017-06-292826, Consejo Nacional de Ciencia y Tecnología). NHR-M and IH-C (100429633-VIEP2021, Vice-Rectory for Research and Postgraduate Studies [VIEP]). Alan Rodríguez Enríquez (BUAP: 219470378) was supported by CONACYT (2019-000037-02NACF). The funders had no role in study design, data collection and analysis, decision to publish, or preparation of the manuscript.

**Competing interests:** The authors have declared that no competing interests exist.

## Introduction

The Porcine Epidemic Diarrhea Virus (PEDV) is the causative agent of porcine epidemic diarrhea, a highly contagious disease that affects farm animals. The disease is diagnosed by clinical symptomatology such as vomiting, diarrhea, dehydration, and anorexia. The infection affects pigs of all ages, with piglets less than one week old being the most affected (100% morbidity and mortality). Although the virus does not affect human health, it causes significant economic losses to the pig breeding industry [1]. Current treatment is primarily aimed at diarrhea controlling. Control strategies have been oriented to the establishment in farms of rigorous biosecurity programs, adequate zootechnical management and vaccination [2]. Although these strategies have been effective in controlling disease, specific treatments are needed, which would require molecular knowledge of each stage of the viral replication cycle, as well as information on the biochemical characteristics of all the viral proteins. PEDV belongs to the order Nidovirales, family *Coronaviridae*, subfamily *Coronavirinae*, genus *Alphacoronavirus*. Its genome is a single-stranded, positive-sense RNA molecule that encodes 16 non-structural proteins (nsp 1–16) and 5 structural proteins, S, M, N, E and ORF3 [2]. Coronavirus M proteins share the same structural characteristics: three TM domains with N-exo-C-endo orientation flanked by a short glycosylated $NH_2$-terminal domain on the virion surface and a long carboxy-terminal globular domain within the virion [3, 4]. The 3D structure of the PEDV M protein is still unknown. The functions described for the PEDV M protein include assembly and budding of virions, inhibition of innate immune response by blocking type I and III interferon pathways in infected cells and induction of protective antibodies in pigs [5–9]. The M protein performs its functions by interacting with other viral proteins. It is known that M proteins interact with each other and with spike and nucleocapsid proteins during virus assembly. Interestingly, M-M interactions form the overall scaffold for the viral envelope [3]. The indispensable role played by the M protein in the viral replication cycle can be explained in part by the lower genetic variability of the M gene compared to the S gene [10]. Therefore, the PEDV M protein could be a good candidate for developing specific antiviral drugs and as a target for vaccine design. The three-dimensional structure of the PEDV M protein has not been reported yet. The structure of some crystallized proteins of PEDV can be found in the PDB database, including the S protein [11], 3CLpro protease [12], NSP9 [13], papain-like protease (PDB: 6NOZ) and NSP1 (PDB 5XBC). Due to the importance of the M protein in the viral infection cycle, the present work proposes a 3D model of the M protein generated using the Robetta protocol. An immunoinformatics analysis was also carried out to determine the potential antigenic regions of the M protein.

## Material and methods

### Sequences

For the generation of the 3D model of the M protein, the sequence of the M protein of the PEDV CV777 virulent strain (Uniprot ID: P59771) (226 residues) and the M protein from strain PEDV/MEX/MICH/01/2013 (GenBank number: MH006957.1) isolated in Michoacán, Mexico, called 2013MMV were used. Different sequences of the M protein were retrieved from the NCBI Taxonomy Browser page (https://www.ncbi.nlm.nih.gov/Taxonomy/Browser/wwwtax.cgi). The sequence of the SARS-CoV-2 M protein of the AlphaFold and Feig lab models corresponds to Uniprot ID: P0DTC5.

### Cloning sequence of the M gen and phylogenetic analyses

The nucleotide sequence of the 2013MMV was subcloned in the vector pPICZαB from the vector pETSumo and it was called 2013MMV. The recombinant DNA was sequenced, 225 amino

acids were obtained by *in silico* translation, Met residue number 1 belongs to the cloning vector. Phylogenetic analyses were conducted in MEGA7 using the Neighbor-joining method [14, 15] using 87 sequences including the two amino acid sequences aforementioned and others from different strains isolated in the United States, Europe and Asia. Evolutionary distances were calculated using the Poisson correction method [16] and are shown as the number of amino acid substitutions per site.

## Programs used to generate 3D protein models and evaluation

The 3D model of the PEDV M protein was generated using Robetta (comparative modeling). CAMEO project (https://www.cameo3d.org/) [17, 18]. The 3D models generated in Robetta were constructed from several templates by Dali server (http://ekhidna2.biocenter.helsinki.fi/dali/) [19]. The 3D M protein models were placed in the membrane using the OPM (https://opm.phar.umich.edu/) by PPM server (http://opm.phar.umich.edu/server.php) to infer its spatial position in the membrane [20]. Images were visualized with FirstGlance in Jmol by PPM server [21]. The quality of the generated models was evaluated using the online servers of QMEAN and QMEANDisCo (https://swissmodel.expasy.org/qmean) [22, 23], PROCHECK (https://saves.mbi.ucla.edu/) [24], MolProbity (http://molprobity.manchester.ac.uk/) [25] and ProSA-web (https://prosa.services.came.sbg.ac.at/prosa.php) [26].

## SARS-CoV-2 M protein models

The 3D models of the SARS-CoV-2 M protein made by Feig lab (https://feig.bch.msu.edu/) and AlphaFold (https://deepmind.com/research/open-source/computational-predictions-of-protein-structures-associated-with-COVID-19) were used to compare the characteristics of these models with the PEDV M protein model generated by Robetta.

## Superimposition of 3D protein models

The superimposition of 3D models and estimation of RMSD values was carried out using the UCSF Chimera program (https://www.cgl.ucsf.edu/chimera/) [27].

## Continuous and discontinuous B-cell epitope prediction

Continuous epitopes in the PEDV M protein were identified using the tools of Immune Data Base (https://www.iedb.org/), Bepipred Linear Epitope Prediction 1.0 and 2.0 http://tools.iedb.org/bcell/). Bepipred 2.0 [28] was used with a threshold of 0.5 (sensitivity = 0.58564, specificity = 0.57158). The Bepipred 1.0 Linear Epitope Prediction tool [29] was used with a threshold of 0.35 (sensitivity = 0.49, specificity = 0.75). The predicted epitopes were evaluated using Chou & Fasman Beta-Turn Prediction, Emini Surface Accessibility Prediction, Kolaskar and Tongaonkar Antigenicity, and Parker Hydrophilicity Prediction with a threshold of 1.0. The conservation analysis was carried out using the IEDB Epitope Conservancy Analysis tool (http://tools.iedb.org/conservancy/) [30]. Discontinuous B-cell epitopes were predicted using the online servers DiscoTope (http://tools.iedb.org/discotope/) [31] and ElliPro (http://tools.iedb.org/ellipro/) [32]. ElliPro with a minimum score of 0.5 and a maximum distance of 6 Å. Regarding DiscoTope (version 1.1), a threshold of -7.7 (sensitivity = 0.47, specificity = 0.75) was used.

## Prediction of cytotoxic T-cell epitopes

Cytotoxic T-cell epitopes were identified using the NetMHCpan 4.1 server [33]. The swine leucocyte alleles (SLA-1:0101; SLA-1:0401; SLA-1:0801) were used, those alleles are widely

distributed in swine populations [34, 35]. The search for peptides covered all nonamers, using 0.5% as a threshold for strong binding and 2% for weak binding.

## Results

### Conservation of the amino acid sequence of PEDV M proteins

To generate the 3D model of PEDV M protein, the amino acid sequence was selected based on a comparison of amino acid sequences from different strains using the M protein sequence of the CV777 prototype virulent strain as a reference. A high degree of conservation was observed among 87 PEDV strains from Mexico, United States of America, Europe, and Asia. The identity between the different sequences analyzed was recorded in the range of 97.35 to 99.56% corresponding to the minimum and maximum identity values related to the CV777 M protein. The amino acid sequences of M protein were clustered into three groups related to G2a, G2b, G1 strains of PEDV (Fig 1). Thus, the high identity value between the compared sequences allowed the selection of the M protein of the PEDV CV777 strain (called M protein CV777 in this work, UniProt ID P59771) to build the 3D model.

### 3D modelling of the PEDV M protein

The predicted 3D model of the PEDV M protein was done using Robetta server on the CAMEO project (Continuous Automated Model EvaluatiOn) [18] using the RosettaCM comparative modeling [36]. The model 1 of the CV777 M protein was selected based on its QMEAN value of -0.86 and its predicted protein architecture, including the transmembrane and globular domains. Also, a 3D model of the M protein from PEDV/MEX/MICH/01/2013 (GenBank number: MH006957.1) called 2013MMV was predicted in this work. Like the CV777 strain, the model 1 was selected for the further analysis. 2013MMV is a Mexican strain isolated in 2013 in Michoacán, México; it has 6 amino acid differences compared to the CV777 M protein, is one of the strains isolated in cell culture that we keep under guard and that was genetically characterized in its entirety by our research group. Fig 2A and 2B show the predicted 3D models of the CV777 M protein and the 2013MMV M protein, respectively. Both models have a short N-terminal ectodomain, three α-helices corresponding to the three successive transmembrane domains, and a long globular C-terminal domain mainly formed by antiparallel β-strands on the inside of the virion. The 3D models generated in Robetta were constructed from various 3D structures deposited in the PDB and were selected by the Dali server, with the SARS-CoV-2 protein ORF3 (6xdc) showing the greatest structural similarity. Then, the 3D models of the CV777 and 2013MMV M proteins were superimposed using the UCSF Chimera program. The main differences were observed in the last 26 amino acids of the 2013MMV M protein on the globular C-terminal domain, which were folded in different spatial orientations. The RMSD value was 12.195 Å (Fig 2C). The theoretical positioning of the CV777 M protein (Fig 2D) and 2013MMV M protein (Fig 2E) in a lipid bilayer was analyzed using the PPM server in OPM. Both 3D M protein models were included in the membrane by three alpha helical transmembrane segments. The general positioning and orientation of both proteins in the lipid bilayer is maintained, but the spatial orientation of the 2013MMV M protein changes in the extracellular region, in the arrangement of the alpha helices in the transmembrane as well as in the globular region. To the 3D model of CV777 M protein, PPM server calculated a value of depth/hydrophobic thickness of 18.3 ± 2.0 Å with a $\Delta G_{transfer}$ to the membrane of -15.1 kcal/mol and a tilt angle of 13±7˚. Similarly, a value of depth/hydrophobic thickness of 20.8 ±1.9 Å, with a $\Delta G_{transfer}$ of -18.8 kcal/mol and tilt angle de 25±0˚ was calculated for the 3D model of 2013MMV M protein.

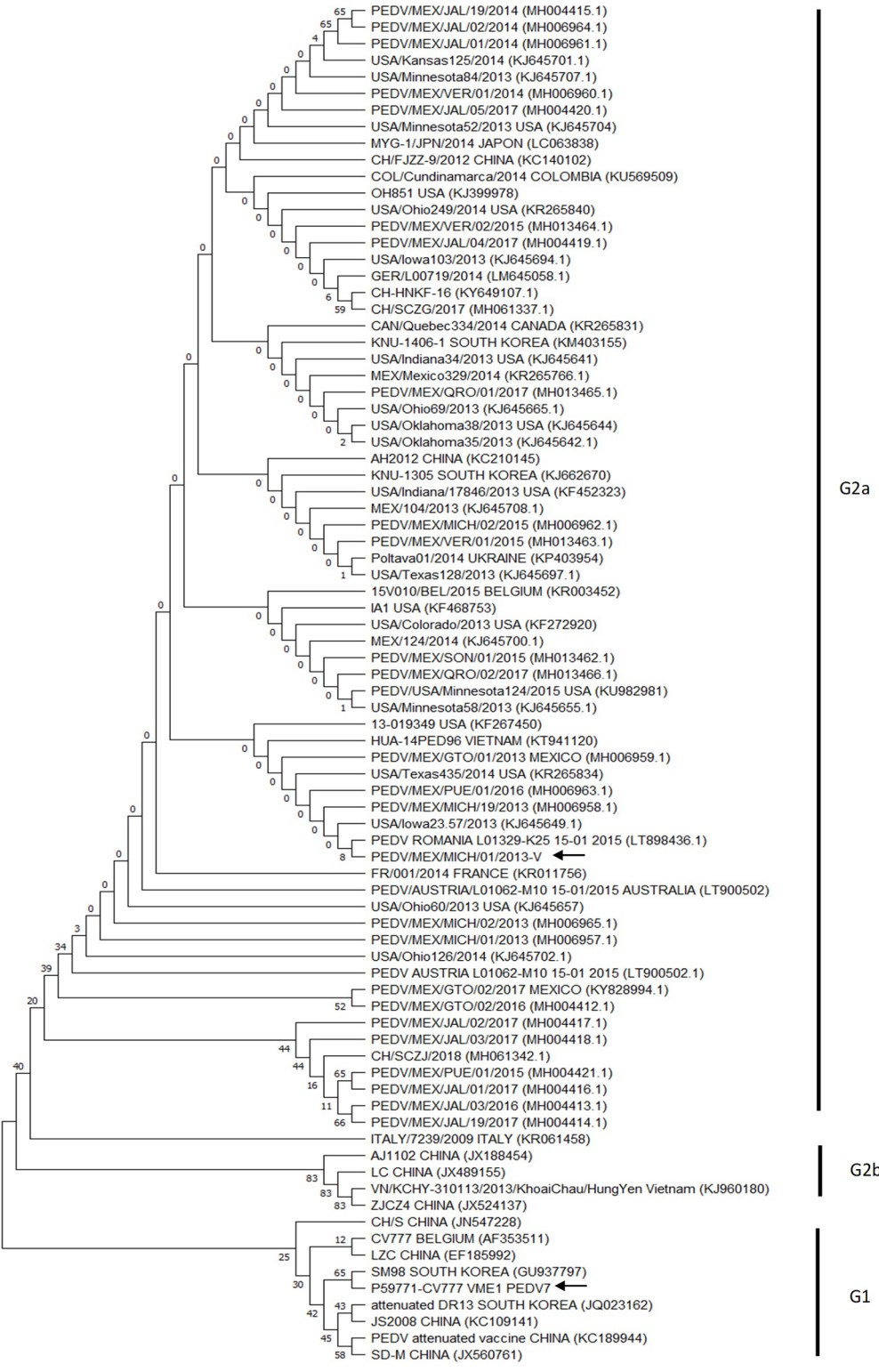

**Fig 1. Phylogenetic tree built with 87 sequences of the M protein of the PEDV from Mexico, United States of America, Europe and Asia.** The amino acid sequences of M protein were clustered into three groups related to G2a, G2b, G1 strains of PEDV. The arrows signal the two sequences chosen to generate the 3D model of the M protein. The evolutionary history was inferred using the Neighbor-Joining method. Details and references are provided in Material and Methods section.

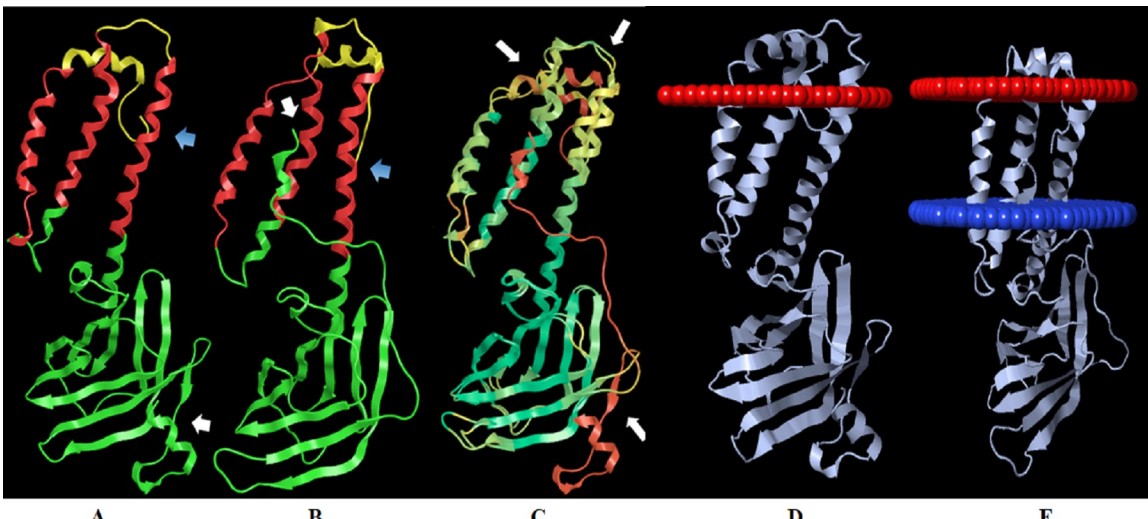

**Fig 2. Predicted 3D models of the CV777 M and 2013MMV M proteins, generated with Robetta.** A) CV777 M protein model. B) 2013MMV M protein model. The transmembrane region is shown in red. The regions outside the viral membrane are shown in yellow. The globular region is shown in green. The blue arrows point to the N-terminal end and the white arrows point to the C-terminal end. C) Overlapping of the 3D models of CV777 M and 2013MMV M proteins. The red areas correspond to the highest RMSD value, while the green areas correspond to the lowest RMSD value. The white arrows indicate the sites with the highest RMSD, corresponding to the loops and to the N- and C-terminal ends. D) 3D model of CV777 M protein on a theoretical lipid layer obtained from the OPM server. The program added red pseudoatoms to mark the extracellular hydrophobic boundary of the lipid bilayer. E) 3D model of 2013MMV M protein on a theoretical lipid bilayer obtained from the OPM server. In this case, the server added red and blue pseudoatoms to mark the extracellular and intracellular hydrophobic boundaries of the lipid bilayer, respectively.

## Validation of the predicted 3D model of the M protein

The predicted 3D models of the CV777 M protein and the 2013MMV M protein were evaluated with different programs. QMEANDisCo (a composite scoring function to derive the absolute quality of the model) values were 0.59 and 0.57, respectively. Global score must be in a range must be in a range between 0 and 1 with one being good. The PROCHECK program determined that the models of CV777 M protein and 2013MM M protein had values of 89.2% and 88.1% of phi and psi angles in the most favored regions of the Ramachandran plot, respectively. According to the PROCHECK server, a good model must contain more than 90% of the phi and psi angles in the most favored regions. But both values are close to 90%. The evaluation of the 3D models of CV777 M protein and 2013MMV M protein in MolProbity yielded scores of 1.34 and 1.37 respectively, indicating an acceptable stereochemistry. Finally, the 3D models were evaluated in ProSA web, CV777 M protein model (Fig 3A and 3B) and the 2013MMV M protein model (Fig 3C and 3D) had Z (overall quality score) values of -5.26 and -5.88, respectively (Fig 3A and 3C). Z-score of the 3D models of M protein is within the range of scores typically found for native proteins of 225 and 226 amino acid residues in length deposited in the PDB. In the plot of residue scores (Knowledge-based energy) of the 3D models of M protein, the 100 residues of the transmembrane region at the N-terminus have the highest energy level. But none of the amino acids had high positive values and were not penalized by the program (Fig 3B and 3D).

## Comparison of predicted M protein 3D models from PEDV and SARS-CoV-2 M protein models

As a result of the SARS-CoV-2 pandemic, DeepMind (AlphaFold) [37, 38] and Michael Feig's laboratory (Feig lab) [39] generated 3D models of the M protein of the SARS-CoV-2 virus.

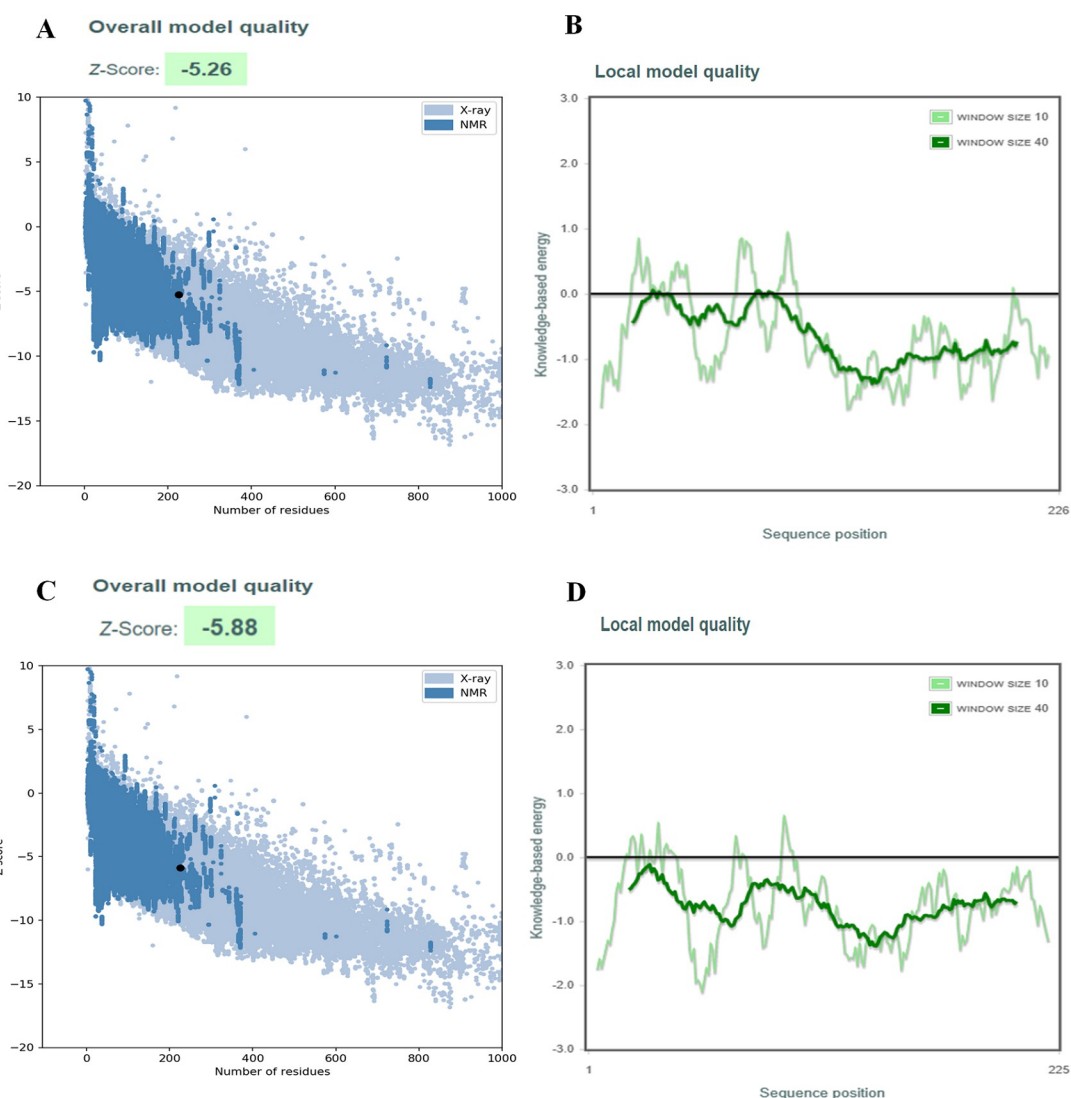

**Fig 3. Evaluation of the overall model quality of 3D models of the CV777 M protein and 2013MMV M protein using ProSA web.** A) Z-score plot of the 3D model of CV777 M protein. B) Plot of residue scores (Knowledge-based energy) of the 3D model of CV777 M protein. C) Z-score plot of the 3D model of 2013MMV M protein. D) Plot of residue scores of the 2013MMV M protein. The black dot in A and C represents the 3D model of the M protein.

AlphaFold uses a neural network to predict the distances between pairs of residues in a protein [37]. Feig lab generated initial machine learning-based models using trRosetta. The models were then subjected to molecular-dynamics simulation-based refinement protocol to maximize model accuracy [39]. Although, SARS-CoV-2 M protein has 33% amino acid sequence identity with CV777 M protein and 94% coverage percentage. 3D models of SARS-CoV-2 and PEDV M proteins displayed a similar global folding (S1 Fig). The SARS-CoV-2 M protein models were evaluated with the programs used with the PEDV M proteins. Then, the results were compared with those obtained from 3D models of the CV777 and 2013MMV M proteins generated by Robetta (S1 Table). The quality parameters of the PEDV M protein models was comparable with those of the validated models from SARS-CoV-2 M protein. To analyze the differences in the theoretical 3D structure of the M proteins, the models of the M proteins from PEDV and SARS-CoV-2 were superimposed. The RMSD values between the CV777 M protein and

SARS-CoV-2 M protein from AlphaFold (S1A Fig) and the CV777 M protein and SARS-CoV-2 M protein from Feig lab models were 6.0 and 14.1 Å, respectively (S1B Fig). The RMSD values between 3D model of 2013MMV M protein with the SARS-CoV-2 M protein model from AlphaFold (S1C Fig) and with SARS-CoV-2 M protein model from Feig lab (S1D Fig) were 8.120 and 15.716 Å, respectively. While the superimposition of the SARS-CoV-2 M protein models from AlphaFold and Feig lab showed an RMSD of 12.6 Å (S1E Fig). The main difference between all the models was observed in the α-helix on the transmembrane region and the spatial arrangement of the globular region. The data obtained demonstrate that the 3D models of CV777 and 2013MMV M proteins generated by Robetta have good overall quality similar to the quality of the models of the SARS-CoV-2 M protein obtained by AlphaFold and Feig lab.

**Identification of linear B cell epitopes in the PEDV M protein model.** The epitopes in the predicted 3D models of the PEDV M protein were analyzed. The B cell epitopes of the CV777 M and 2013MMV M proteins were predicted using servers BepiPred 1.0 (linear epitope prediction) and Bepipred 2.0 (sequential B cell epitope predictor). Both programs predicted four peptides in the CV777 M protein model; the B cell peptides RSVNASSGTG and KHGDY-SAVSNPSAVLT were predicted by both servers. The two peptides were above the threshold level of 1.0 (Table 1). In the 2013MMV M protein model, 3 and 4 peptides were predicted by both servers, respectively. The KHGDYSAVSNPSSVLTD epitope was above the threshold level of 1.0 and was predicted by both servers. The peptides identified in the 3D models of the

**Table 1. Continuous B-cell epitopes of CV777 M protein.**

| Continuous B-cell epitopes | Position | Length | Chou | Emini | Karplus | Kolaskar | Parker | Conservation % | Identity % Minimum-maximum |
|---|---|---|---|---|---|---|---|---|---|
| **CV777 M protein model** | | | | | | | | | |
| **Bepipred 1.0** | | | | | | | | | |
| MSNGSIPV | 1–8 | 8 | 1.217 | 1.000 | 1.065 | 0.999 | 2.264 | 90.80 | 37.50–100.00 |
| SFNPETDAL | 111–119 | 9 | 1.108 | 1.000 | 1.051 | 0.951 | 3.781 | 100.00 | 100.00–100.00 |
| RSVNASSGTG | 185–194 | 10 | 1.207 | 1.000 | 1.038 | 0.993 | 4.561 | 94.25 | 90.00–100.00 |
| **KHGDYSAVSNPSAVLT** | **202–217** | **16** | **1.133** | **1.000** | **1.004** | **1.057** | **2.986** | 16.09 | 93.75–100.00 |
| **Bepipred 2.0** | | | | | | | | | |
| SIPVDEVIEHLRNWNF | 5–20 | 16 | 0.863 | 1.00 | 0.973 | 1.046 | 0.497 | 12.64 | 56.25–100.00 |
| YKVATGVQVSQL | 154–165 | 12 | 0.910 | 1.000 | 0.989 | 1.106 | 1.960 | 100.00 | 100.00–100.00 |
| **GRVGRSVNASSGTG** | **181–194** | **14** | **1.142** | **1.000** | **1.026** | **1.011** | **3.807** | 93.10 | 92.86–100.00 |
| **KHGDYSAVSNPSAVLTDSEKV** | **202–222** | **21** | **1.078** | **1.000** | **1.006** | **1.055** | **2.883** | 14.94 | 90.48–100.00 |
| **2013MMV M protein model** | | | | | | | | | |
| **Bepipred 1.0** | | | | | | | | | |
| SFNPETDAL | 110–118 | 9 | 1.108 | 1.000 | 1.051 | 0.951 | 3.781 | 100.00 | 100.00–100.00 |
| RSVNASSGTG | 184–193 | 10 | 1.207 | 1.000 | 1.038 | 0.993 | 4.561 | 94.25 | 90.00–100.00 |
| **KHGDYSAVSNPSSVLTD** | **201–217** | **17** | **1.172** | **1.000** | **1.019** | **1.056** | **3.169** | 83.91 | 94.12–100.00 |
| **Bepipred 2.0** | | | | | | | | | |
| VDEVIQHLRNWNF | 7–19 | 13 | 0.891 | 1.00 | 0.952 | 1.042 | 0.086 | 78.16 | 61.54–100.00 |
| YKVATGVQVSQL | 153–164 | 12 | 0.910 | 1.000 | 0.989 | 1.106 | 1.960 | 100.00 | 100.00–100.00 |
| **RVGRSVNASSGTG** | **181–193** | **13** | **1.153** | **1.000** | **1.026** | **1.007** | **3.965** | 94.25 | 92.31–100.00 |
| **KHGDYSAVSNPSSVLTDSEKV** | **201–221** | **21** | **1.129** | **1.000** | **1.023** | **1.052** | **3.176** | 82.76 | 90.48–100.00 |

Gray shading indicates epitopes with values above 1 in all parameters.

PEDV M proteins were used to perform a conservation analysis (%) with 31 M protein sequences from different PEDV strains isolated in Mexico and different strains isolated in the United States and other countries. A high level of conservation of the epitopes identified in the M proteins of 2013MMV and CV777 was observed among the analyzed sequences. However, the sequences of some epitopes from the 2013MMV M protein were more highly conserved than those from the CV777 M protein (Table 1).

**Prediction of discontinuous B-cell epitopes from the PEDV M protein model.** The prediction of discontinuous epitopes from M protein 3D models from PEDV and SARS-CoV-2 was performed using the ElliPro and DiscoTope servers. ElliPro predicted 6 epitopes in the CV777 M protein model, 4 in the 2013MMV M protein model, 4 in the SARS-CoV2 M protein model from Alpha fold and 5 in the SARS-CoV2 M protein model from Feig lab (Fig 4) (S2 Table). The DiscoTope program predicted 15 residues in the CV777 model and 14 residues in the 2013MMV M protein model with potential to form an epitope, while 14 and 24 residues were predicted in the SARS-CoV-2 M protein model from AlphaFold and the SARS- CoV-2 M protein model from Feig lab, respectively (Fig 5) (S3 Table). In the Figs 4 and 5, the predicted epitopes were located in protein regions with similar folding. In both figures the colored regions correspond to discontinuous epitopes, ordered from the highest (red) to the lowest score (orange).

## Prediction of T-cell epitopes for the PEDV M protein

Prediction of T-cell epitopes was performed using the NetMHCpan 4.1 server (DTU Bioinformatics). Three alleles were used for the prediction, SLA-1: 0101, SLA-1: 0401 and SLA-1:0801.

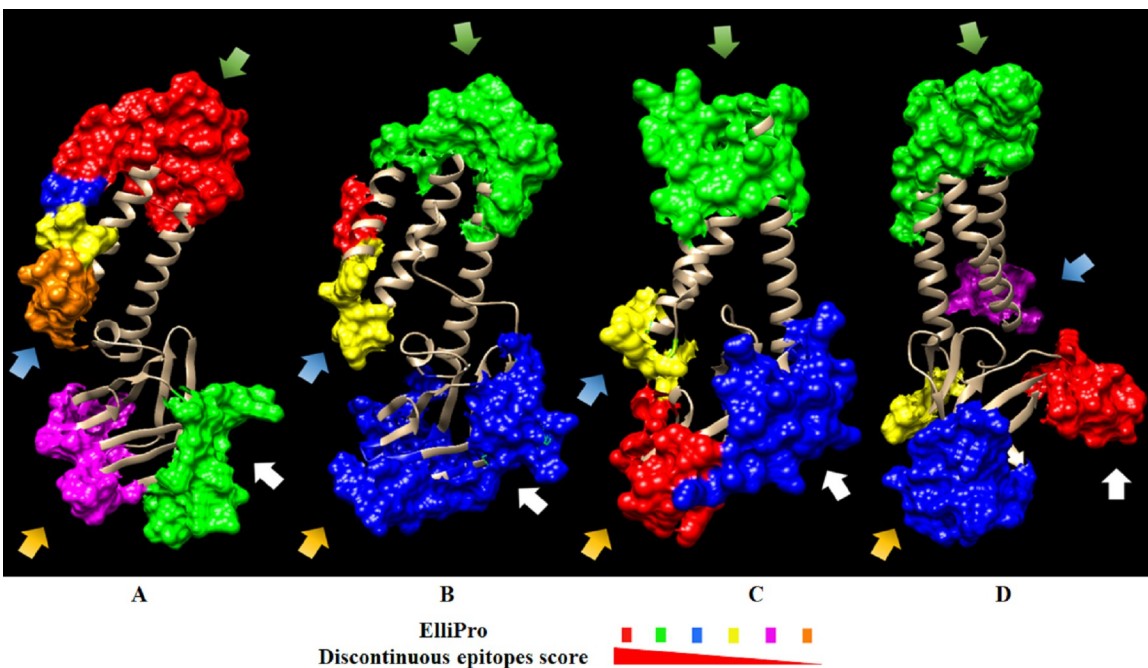

**Fig 4. Discontinuous epitopes predicted in ElliPro from CV777, 2013MMV and SARS-CoV-2 M protein models.** A) CV777 M protein. B) 2013MMV M protein. C) SARS-CoV-2 M protein from AlphaFold. D) SARS-CoV-2 M protein from Feig lab. Predicted discontinuous epitopes are highlighted in the 3D model with different colors to represent the score. The epitope with the highest score is represented in red up to the lowest value in orange. The score decreases from the epitopes marked in green, blue, yellow, and pink. Arrows with the same color indicate discontinuous epitopes located in equivalent regions in the models. Colored residues in each epitope are indicated in S2 Table.

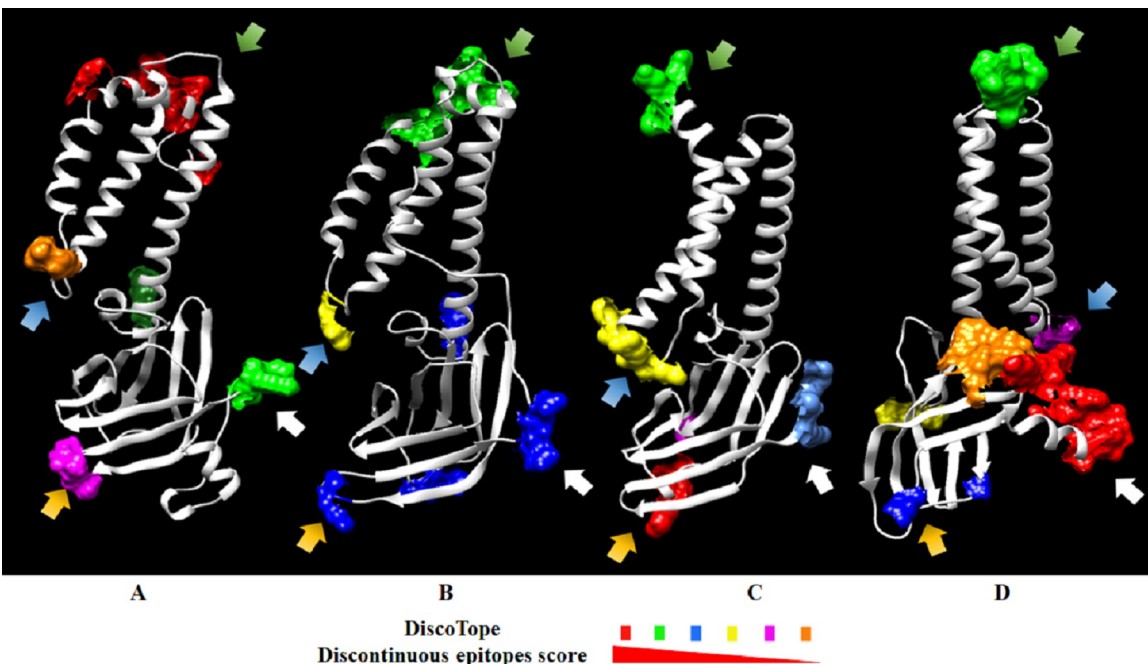

**Fig 5. Discontinuous epitopes predicted in DiscoTope from CV777 and 2013MMV M protein and SARS-CoV-2 protein M models.**
A) CV777 M protein. B) 2013MMV M protein. C) SARS-CoV-2 M protein from AlphaFold. D) SARS-CoV-2 M protein from Feig lab.
Predicted discontinuous epitopes are highlighted in the 3D model with different colors to represent the score. The epitope with the
highest score is represented in red up to the lowest value in orange. The score decreases from the epitopes marked in green, blue, yellow
and pink. Arrows with the same color indicate discontinuous epitopes located in equivalent regions in the models. An additional color
(light blue) show residues that were not predicted in ElliPro. Colored residues in each epitope are indicated in S3 Table.

In the CV777 M protein, 40 weak-binding and 14 strong-binding peptides were predicted
(Table 2). In the 2013MMV M protein, 36 weak-binding and 16 strong-binding peptides were
predicted. The immunogenicity of the strong binding peptides of both M proteins was pre-
dicted using the IEDB class 1 immunogenicity server, which yielded 10 epitopes, 2 of them
repeated in different alleles. The study provides theoretical data on the potential of the M pro-
tein to be considered as an immunogen in the development of a new vaccine for the control of
PEDV.

## Discussion

The porcine industry is affected each year by the loss of pigs due to viral infections, with por-
cine epidemic diarrhea being one of the most worrying diseases for producers. The disease,
which is highly contagious, is caused by PEDV. PEDV is an etiological agent associated with
infection outbreaks throughout the world, including Mexico. Although this disease is dealt
with using biosecurity programs, adequate zootechnical management practices and vaccines,
the occurrence of mutations in the viral genome motivates the continuous search for new con-
trol strategies. This requires molecular knowledge of the potential viral triggers of the host's
immune response. The M protein fulfills several important viral functions in different regions
of the viral particle structure. Knowledge of these functions could help develop new strategies
to control the disease. However, structure of the M protein of PEDV is unknown. Thus, the
present work generated a 3D model of the M protein and compared it to validated models of
SARS-CoV-2. The potential regions inducing an immune response were evaluated in the 3D
model; 4 promising sites were found that should be subjected to further study. The 3D model

was constructed based on the amino acid sequence of the M protein of CV777 and on that of a Mexican strain isolated in Michoacán, Mexico, showing changes of amino acids compared to the prototype. This could help to determine if the 3D structure of the protein is significantly affected when changes in the amino acid sequence occur, especially in the globular region of the protein where the residues associated with protein functions are found. To confirm that the amino acid sequences selected to obtain the 3D model represent the M protein of PEDV, a comparison was made with strains from different outbreaks around the world. The sequences were clustered into three groups related to G2a, G2b, G1 strains of PEDV. The analysis included the genogroups reported for the PEDV. There are different genogroups of PEDV, G1 and G2 (classic strain and variant strains), which can be divided into G1a and G1b, G2a and G2b. The G1a genogroup includes classic strains such as the virulent CV777 prototype strain and other ancient PEDV strains. The G1b genogroup contains strains isolated in Asia with deletions in different genes, mainly in the genes encoding to the nsp3 and ORF3 proteins [40]. Genogroup G2 differs from G1 mainly due to amino acid changes in the NTD domain of protein S. Within the genogroup G2a there are circulating Asian strains reported in reemerging cases of PEDV and in the genogroup G2b there are circulating strains related to outbreaks in the United States and other parts of the world [41]. The data obtained support the construction of the 3D model from the sequences of CV777 strain and the Mexican sequence. The present work proposes for the first time a predicted 3D model of the M protein of PEDV. The model was generated by Robetta using comparative modelling, using the ORF3 protein of SARS-CoV-2 as one of the main modeling templates. The structure of ORF3 (PDB: 6XDC) was solved in 2021 using Cryo-EM and has the same topology in the globular endodomain M as the PEDV M protein [42]. The 3D model of the M protein shown in Fig 2 shows a topology consistent with the architecture reported for the coronavirus M protein. It is organized into two transmembrane and globular domains, with 3 α-helices that would cross the membrane, with loop regions inside and outside the viral envelope and a globular domain formed mainly by antiparallel β-strands. Interestingly, the 3D model preserves the topology reported in Uniprot for the CV777 M protein: virion surface (positions 1–11), transmembrane (12–32, 21 residues of length), intravirion (33–41, 9 residues), transmembrane (42–62, 21 residues), virion surface (63–75, 13 residues), transmembrane (76–96, 21 residues), intravirion (97–226, 130 residues) (https://www.uniprot.org/uniprot/P59771).

Analysis of the positioning in the lipid bilayer of the theoretical 3D structure of M proteins by the PPM server showed the overall conservation of the theoretical topology, but the amino acid changes in the 2013MMV M protein change the spatial arrangement in the extracellular region, the tilt angle of the transmembrane alpha helices and the globular region. The PPM server added red and blue pseudoatoms to mark the extracellular and cytoplasmic hydrophobic boundaries of the lipid bilayer, respectively (Fig 2D and 2E). Although PPM was only able to calculate both membrane limits in the positioning of the 2013MMV M protein but not in the CV777 M protein, in the last one only the extracellular limit was calculated. This interesting finding will be analyzed later by *in silico* and experimental studies to evaluate the potential effect on the recognized functions of the M protein in the conformation of the virion and on some processes in cells infected by PEDV.

The M protein is the most abundant structural protein in the viral envelope of PEDV and other coronaviruses and it is known to play a central role in the assembly of the virus. Homodimerization of the M protein through the transmembrane region has been reported [43]. In Mouse Hepatitis Virus (MHV), the interaction of the M protein with the E protein is the minimum requirement for the assembly of the viral particle [43], Kuo and Masters reported an interaction between the M and N proteins of MHV through the C-terminal end of the M protein [44]. Interestingly, it was shown that, in the absence of the N protein, the MHV M protein

**Table 2. Continuous T cell epitopes of CV777 and 2013MMV M protein.**

**Allele SLA-1:0101**

| Weak binding peptides | %Rank EL | %Rank BA | Affinity Nm | Strong binding peptides | %Rank EL | %Rank BA | Affinity Nm | Class 1 immunogenicity |
|---|---|---|---|---|---|---|---|---|
| GSIPVDEVI | 1.379 | 1.825 | 7527.49 | PVDEVIEHL | 0.135 | 1.608 | 7076.59 | 0.37683 |
| ILWPLVLAL** | 1.278 | 1.294 | 6391.21 | LGAPTGVTL** | 0.165 | 0.384 | 3530.72 | 0.12876 |
| SLFDAWASF** | 0.795 | 0.464 | 3876.20 | APTGVTLTL** | 0.360 | 1.183 | 6126.86 | 0.13676 |
| SFNPETDAL** | 0.938 | 2.037 | 7930.56 | SQLPNFVTV** | 0.055 | 0.048 | 1310.84 | 0.14666 |
| TDALLTTSV** | 1.285 | 1.210 | 6194.39 | PVDEVIQHL* | 0.105 | 1.758 | 7386.78 | 0.19457 |
| QVSQLPNFV** | 1.499 | 2.900 | 9428.25 | AVSNPSAVL | 0.218 | 0.411 | 3653.72 | -0.1696 |
| TVAKATTTI** | 0.505 | 0.761 | 4916.88 | SAVSNPSSV* | 0.476 | 0.682 | 4683.27 | -0.40609 |
| ASSGTGWAF** | 0.787 | 0.490 | 3991.62 | AVSNPSSVL* | 0.245 | 0.746 | 4875.56 | -0.34224 |
| YSAVSNPSA | 1.404 | 0.869 | 5261.96 | | | | | |
| SAVSNPSAV | 0.621 | 0.580 | 4323.47 | | | | | |
| AVLTDSEKV | 1.561 | 2.739 | 9167.12 | | | | | |
| VLTDSEKVL | 1.988 | 8.192 | 15472.46 | | | | | |
| TDSEKVLHL** | 0.946 | 8.053 | 15354.38 | | | | | |
| DSEKVLHLV** | 1.600 | 5.414 | 12759.76 | | | | | |
| SAFLYGVKM* | 1.695 | 2.275 | 8362.81 | | | | | |
| SVLTDSEKV* | 1.383 | 2.249 | 8314.10 | | | | | |
| VLTDSEKVL* | 1.988 | 8.192 | 15472.46 | | | | | |

**Allele SLA-1:0401**

| Weak binding peptides | %Rank EL | %Rank BA | Affinity Nm | Strong binding peptides | %Rank EL | %Rank BA | Affinity Nm | Class 1 immunogenicity |
|---|---|---|---|---|---|---|---|---|
| PVDEVIEHL | 1.470 | 4.053 | 4846.79 | VLQYGHYKY** | 0.211 | 0.270 | 238.88 | -0.10699 |
| EVIEHLRNW | 1.011 | 1.352 | 1420.59 | SLFDAWASF** | 0.189 | 0.164 | 148.60 | 0.2429 |
| TILLVVLQY** | 0.972 | 2.606 | 3009.49 | ASSGTGWAF** | 0.226 | 0.126 | 110.35 | 0.2599 |
| SVFLYGVKM | 1.859 | 2.498 | 2882.55 | AVSNPSSVL* | 0.481 | 0.763 | 751.51 | -0.34224 |
| ASFQVNWVF** | 1.431 | 1.567 | 1700.35 | | | | | |
| QVNWVFFAF** | 1.889 | 0.978 | 987.97 | | | | | |
| RLWRRTHSW** | 0.689 | 0.582 | 552.91 | | | | | |
| LTLLSGTLF | 0.953 | 0.933 | 936.80 | | | | | |
| TVAKATTTI** | 1.186 | 1.485 | 1599.22 | | | | | |
| SSGTGWAFY** | 0.863 | 0.382 | 339.85 | | | | | |
| AVSNPSAVL | 0.553 | 0.692 | 679.83 | | | | | |
| LTDSEKVLH** | 0.794 | 1.551 | 1682.18 | | | | | |
| VQVSQLPNF* | 1.454 | 2.016 | 2249.84 | | | | | |
| PVDEVIQHL* | 0.726 | 2.470 | 2842.62 | | | | | |
| EVIQHLRNW* | 1.597 | 2.071 | 2314.73 | | | | | |

**Allele SLA-1:0801**

| Weak binding peptides | %Rank EL | %Rank BA | Affinity Nm | Strong binding peptides | %Rank EL | %Rank BA | Affinity Nm | Class 1 immunogenicity |
|---|---|---|---|---|---|---|---|---|
| EVIEHLRNW | 0.837 | 1.907 | 6253.17 | TILLVVLQY** | 0.409 | 1.346 | 4685.44 | -0.1274 |
| HLRNWNFTW** | 1.228 | 0.856 | 3057.97 | VLQYGHYKY** | 0.126 | 0.214 | 850.54 | -0.10699 |
| LVVLQYGHY** | 1.508 | 0.767 | 2751.72 | SLFDAWASF** | 0.039 | 0.011 | 117.99 | 0.24329 |

(*Continued*)

**Table 2.** (Continued)

| | | | | | | | | |
|---|---|---|---|---|---|---|---|---|
| SVFLYGVKM | 0.944 | 1.175 | 4166.01 | RLWRRTHSW** | 0.451 | 0.445 | 1647.27 | 0.14156 |
| ILWPLVLAL** | 1.053 | 0.874 | 3124.42 | VQVSQLPNF** | 0.441 | 0.678 | 2445.83 | -0.28945 |
| ASFQVNWVF** | 0.827 | 0.776 | 2780.60 | ASSGTGWAF** | 0.322 | 0.188 | 762.6 | 0.2599 |
| QVNWVFFAF** | 1.312 | 0.513 | 1889.30 | | | | | |
| TLMLWIMYF** | 1.126 | 0.569 | 2071.38 | | | | | |
| LTLLSGTLF | 0.969 | 0.649 | 2342.52 | | | | | |
| SQLPNFVTV** | 0.864 | 1.650 | 5612.88 | | | | | |
| AKATTTIVY** | 0.599 | 1.520 | 5225.76 | | | | | |
| SSGTGWAFY** | 1.186 | 0.635 | 2296.34 | | | | | |
| YVRSKHGDY** | 1.263 | 1.539 | 5282.61 | | | | | |
| AVSNPSAVL | 0.702 | 1.251 | 4399.64 | | | | | |
| SAFLYGVKM* | 1.087 | 1.256 | 4413.65 | | | | | |
| AVSNPSSVL* | 0.645 | 1.440 | 4974.67 | | | | | |
| EVIQHLRNW* | 1.293 | 2.977 | 8876.16 | | | | | |
| IQHLRNWNF* | 1.807 | 2.405 | 7434.73 | | | | | |

* Unique epitopes in 2013MMV M protein.

** Epitopes that appear in CV777 and 2013MMV M protein.

can interact with synthetic RNAs containing a viral packaging signal that favors the specific packaging of RNAs in coronavirus-like particles [45]. A recent study showed that the PEDV M protein can interact with S and E viral proteins, with ORF3 and with 218 host proteins. Some of them are associated with 131 signaling pathways and 10 biological processes [46]. The M protein interacts with the translation initiation factor 3 (eIF3L) whose downregulation significantly increases viral production. The translation initiation factor 3 could act as a negative regulator of PEDV replication [46]. It has been shown that the M protein of three coronaviruses, SARS, MHV and the feline coronavirus (FCoV), adopts 2 conformations, called long and compact, that influence the curvature of the viral envelope. The long conformation (M-long) provides rigidity and uniformity to the viral envelope [47]. It has also been demonstrated that the M protein forms homodimers through contact with region corresponding to the endodomain, and that the M-long conformation interacts mainly with N and S proteins in a stoichiometry of the 8M2: 4N: 1S3 type [47]. The multimerization of the M protein is a conserved process among coronaviruses [47]. Recently, Lim Heo and Michael Feig generated models of homodimers of the M protein, including 4 homodimers of the M protein interacting with a trimer in the transmembrane region of the S protein of SARS-CoV-2. The models are deposited in GitHub (https://github.com/feiglab/sars-cov-2-proteins) [39]. Knowing the interactions of the M protein, as well as knowing its 3D structure, or even having a good 3D model thereof, would allow making a deep analysis of the intracellular processes this protein carries out, which would in turn allow the design of drugs and other alternatives that could stop the M protein from performing its functions as a method for controlling infections caused by PEDV and other coronaviruses. The lack of efficient control strategies to reduce the spread of the PEDV and, therefore, the death of piglets, keeps alive the interest in the development of new control methods. However, the constantly changing viral genome makes difficult to achieve this goal. Different studies have reported the existence of different recombination events

between PEDV strains [48–51]. A recombination event between a PEDV strain and one of the transmissible gastroenteritis virus (TGEV) was reported in Russia and Italy [52, 53]. A recombination event between a highly virulent PEDV strain and an attenuated PEDV strain has also been reported; this event generated a new virulent strain [54]. Developing an efficient control strategy has been hindered by the occurrence of mutations and recombinations among PEDV strains in the coding region of protein S. Therefore, it is important to study the immunological and functional potential of other structural proteins of the virus. The M protein is a good candidate due to its low genetic variability, its structure, and the functions it might perform. It is particularly interesting to analyze the potential of the M protein to induce the activation of the immune system. It has been shown that in other coronaviruses, such as MHV, the immune system produces monoclonal antibodies directed to the M protein; these antibodies protect mice from viral infection [55]. The immunization of 8 kittens with the recombinant M protein of the feline infectious peritonitis virus (FIPV) allowed 3 of them to survive the viral infection [56]. B-cell epitopes have also been found in the M protein of SARS-CoV as well as the avian infectious bronchitis virus [57, 58]. It has also been experimentally demonstrated that some peptides of the SARS-CoV M protein are highly reactive with sera from the convalescent phase of SARS patients. High titers of antibodies against the M protein were also elicited in immunized mice and rabbits, which suggests that the M protein is a highly immunogenic component in inactivated SARS vaccine preparations [59]. An experimental study identified a B cell epitope in the PEDV M protein that corresponds to the [195]WAFYVR[200] motif [8] covering the C-terminus of M protein. In the present study, the prediction of B cell epitopes of the M protein of PEDV was carried out and although there is agreement in the results for most of the programs used, it is noteworthy that the epitope [195]WAFYVR[200] found by Zhang [8] was not predicted. Some epitopes in the globular region at the C-terminus of M protein were located before or after the [195]WAFYVR[200] sequence. Further research of the immunogenic potential of the M protein could involve expressing the M protein of coronaviruses in eukaryotic and prokaryotic systems to analyze possible changes in the conformation of the protein. It would also be necessary to analyze the contact regions of the M protein with the antibodies produced by pigs immunized with the PEDV M protein [8].

The *in silico* results of the present study suggest that the M protein is a potential candidate for the activation of the cellular immune response against to the PEDV infection. Finally, a high degree of conservation of each of the B and T cell epitopes predicted in this study was also demonstrated by comparing different strains of PEDV circulating in Mexico and other parts of the world.

## Conclusion

Predicted 3D models of the CV777 M and 2013MMV M proteins were obtained. *In silico* identification of continuous and discontinuous B cell epitopes and T cell epitopes on the M protein was also performed. The comparison between the 3D models of PEDV M proteins obtained in this study and the M protein models from SARS-CoV-2 obtained by the AlphaFold group and the Feig lab showed a high conservation of the 3D structure in both viruses, as well as the presence of discontinuous epitopes in the same regions in both proteins. The 3D model of the PEDV M protein generated in the present study could be a useful tool to carry out further research on the development of vaccines or drugs to control PED.

## Supporting information

**S1 Fig. Superimposition of the 3D M protein models from PEDV and SARS-CoV-2.**
(PDF)

**S1 Table. Comparison of the quality of the 3D models of the M proteins from PEDV and SARS-CoV-2.**
(PDF)

**S2 Table. Prediction of discontinuous epitopes with ElliPro in the 3D M protein models.**
(PDF)

**S3 Table. Prediction of discontinuous epitopes with DiscoTope in the 3D M protein models.**
(PDF)

## Author Contributions

**Conceptualization:** Alan Rodríguez-Enríquez, Irma Herrera-Camacho, Julio Reyes-Leyva, Gerardo Santos-López, José Francisco Rivera-Benítez, Nora Hilda Rosas-Murrieta.

**Formal analysis:** Alan Rodríguez-Enríquez, Nora Hilda Rosas-Murrieta.

**Funding acquisition:** Irma Herrera-Camacho, José Francisco Rivera-Benítez, Nora Hilda Rosas-Murrieta.

**Investigation:** Alan Rodríguez-Enríquez, Gerardo Santos-López, Nora Hilda Rosas-Murrieta.

**Methodology:** Alan Rodríguez-Enríquez, Nora Hilda Rosas-Murrieta.

**Project administration:** Irma Herrera-Camacho, Nora Hilda Rosas-Murrieta.

**Software:** Alan Rodríguez-Enríquez.

**Supervision:** Irma Herrera-Camacho, Julio Reyes-Leyva, Gerardo Santos-López, José Francisco Rivera-Benítez, Nora Hilda Rosas-Murrieta.

**Validation:** Lourdes Millán-Pérez-Peña, Julio Reyes-Leyva, Gerardo Santos-López, José Francisco Rivera-Benítez, Nora Hilda Rosas-Murrieta.

**Visualization:** Alan Rodríguez-Enríquez, Nora Hilda Rosas-Murrieta.

**Writing – original draft:** Alan Rodríguez-Enríquez, Nora Hilda Rosas-Murrieta.

**Writing – review & editing:** Lourdes Millán-Pérez-Peña, Julio Reyes-Leyva, Gerardo Santos-López, José Francisco Rivera-Benítez, Nora Hilda Rosas-Murrieta.

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
