## [Decision Letter · Decision Letter 0]

31 Aug 2021

PONE-D-21-16531

Predicted 3D model of the M protein of Porcine Epidemic Diarrhea Virus and analysis of its immunogenic potential

PLOS ONE

Dear Dr. Rosas-Murrieta,

Thank you for submitting your manuscript to PLOS ONE. After careful consideration, we feel that it has merit but does not fully meet PLOS ONE’s publication criteria as it currently stands. Therefore, we invite you to submit a revised version of the manuscript that addresses the points raised during the review process.

The reviewers raised important points about the manuscript. All points must be addressed in the point-by-point rebuttal letter and in most cases should be incorporated in the manuscript itself. In particular, Reviewer 2 raises a point about the intrinsic inaccuracy of the prediction of T cell epitopes. The authors must provide comments about the limitations of the approach and how it affects the validity of the prediction. As suggested, shwoing a consensus prediction may be helpful. This reviewer also stresses out the validity of Autodock Vina in analyzing protein-peptide interactions. Again, the authors must address the limitations of this approach and provide the explanations required by this reviewer (assigning binding energy, standard deviations and reference test) in the manuscript. The authors should also comment on the suggestion from Reviewer 1 of modeling the M protein in membrane.

In addition to the reviewer comments. a list of comments from the Academic Editor is added below. Please address these points in the manuscript as well. In particular, care must be taken to correct the referencing system, since the current numbering of references in the text and in the bibliography do not match. The accuracy of the references could not be evaluated and will be reviewed in the revised manuscript. In addition, because PLOS One does not provide copy-editing, it is important to revise the manuscript for clarity and typographic errors. I listed below a number of changes to consider in a revised manuscript to improve clarity.

Academic Editor's comments:

1) The referencing system is inaccurate and references numbers in the text do not match references listed in the bibliography. In the text, references are numbered in order of appearance, while in the list, references are numbered alphabetically. Thus numbers do not match. Please use the PLOS One recommended format from Instructions to authors:  References are listed at the end of the manuscript and numbered in the order that they appear in the text. In the text, cite the reference number in square brackets (e.g., “We used the techniques developed by our colleagues [19] to analyze the data”).

When revising the references, please make sure the formatting is consistent. In particular the use of Title Capitalization is inconsistent (e.g .refs 11 and 27, maybe others)

2) Legend of figure 1 contains unformatted references. Due to potential formatting issues, I do not recommend placing references in figure legends. It is better to provide the detailed methodological description in the material and methods section. In the legend, provide a simplified description of the approach and indicate “see material and methods for details and references”.

3) There appears to be a discrepancy between the QMEAN values cited in the text (L131) and in Table 1.  The text mentions -14.90 but the table mentions -14.60. Please correct or clarify.

There is another discrepancy of numbers between line 91 and 336 (percent identity). Please carefully revise numerical data since such discrepancies may negatively reflect on the study.

4) Ensure that figure legends are complete:

- Figure legend 1: indicate the significance of arrows. The text (L93) refers to three groups, clarify how these groups are identifiable from the tree.

- Figure legend 2: although panel C is described in a subsequent paragraph, the complete legend for figure 2 should be kept together (L189-192 added at L157)

5) Ensure that all figures have a sufficient resolution so that they are easily readable when magnified.

6) Typographic errors and minor changes:

L33: … virus (PEDV) is …

L38: interferon pathways

L91: 97.35-99.56%

L95: as a reference

L124: It is clear

L132: remove period before parenthesis

Table S6: (Excl. Gly and Pro)

Table S6: define asterisks

L146: It is mentioned that the most important template was ORF3. It’s not clear how “importance” is defined and evaluated. Pleas clarify. If the different models had different wight in generating the models, it should be indicated in the table.

Figure S7A: for consistency use “Model 1 of 2013MMV M protein”

L207: reference needed for both approaches

Figure S8: provide significance of arrows and colors in legend

L215: 33% amino acid sequence identity

Table S9: PROCHECK

L243: B cell

Table S10: first Row: Parker, Conservation. Hyphenate or adjust column width.

Table S10: 2013MMV first peptide. Position cannot be 110-112.

Figure 5: it would be helpful if the colored residues were identified on the models

Table S13: Remove rows in Spanish for 0401 and 0801.

Table S13: the title of the table refers to epitopes of CV777, but is contains a number of epitopes unique to 2013MMV (*). Modify the title of the table so that it better reflects its content.

Table 3: it’s not clear to me why there is a range of identity. Please clarify.

L304: the data do not really address the “feasibility” od a candidate vaccine, rather, it suggest the potential of the M protein for the development of a candidate vaccine. Please clarify.

Figure 6: since the first B cell epitope is colored in red  inf figure s 4 and 5, it is confusing to color Tcell epitope also in red in figure 6. Make sure that the two types of epitopes are clearly differentiated in this figure.

Figure 6: for consistency and simplification, use the 2013MMV abbreviation rather than the full name

L319: practices

L320: triggers of the host’s

L324: PEDV was unknown

L326: the potential regions inducing an immune response

L328: constructed

L333-335: unclear sentence. Please rephrase

L336: 99.11% identity

L336: the identity numbers are different from the same sentence in line 91. Please explain or correct.

L348: beta-strands

L349: residues in length

L358: remove reference labeled by author names

L362: … 218 host proteins [reference needed]

L363, L368, L372: [reference also needed for these statements]

L376: verify the link (space in the link suggest it is incorrect)

L400: reference format [31] [32] should be [31-32]

L405: reference for Zhang paper

L406: “last figure of the article” unclear which article this refers to. The last figure of this manuscript does not offer a comparison with other epitopes. The discussion comparing the present data and previously known M protein epitopes must be detailed and clarified (L398-399 and L407)

L413: the docking data do not confirm that these epitopes can bind MHC. The data suggest that the epitope can bind MHC. Binding has to be demonstrated before being confirmed.

L416: do you mean “demonstrated by comparing different strains”? please clarify.

L420: B cell and T cell not hyphenated

L420: this sentence could be interpreted that the epitopes were identified functionally. Indicate that these are proposed epitopes identified in silico.

L425: Unclear: “showed to be” appears incorrect. Do you mean “will be”?

L435: corresponds

We look forward to receiving your revised manuscript.

Kind regards,

Claude Krummenacher, PhD

Academic Editor

PLOS ONE

Journal Requirements:

Funding: JFR-B, IH-C and NHRM (FONSEC-SAGARPACONACYT 2017-06-292826).

JFRB, IHC, NHRM 

Grant number: FONSEC-SAGARPACONACYT 2017-06-292826

Consejo Nacional de Ciencia y Tecnología

https://www.conacyt.mx/Convocatorias-fondos-sectoriales-constituidos.html 

NO, The funders had no role in study design, data collection and analysis, decision to publish, or preparation of the manuscript.

Reviewers' comments:

Reviewer's Responses to Questions

**Comments to the Author**

1. Is the manuscript technically sound, and do the data support the conclusions?

Reviewer #1: Yes

Reviewer #2: Partly

2. Has the statistical analysis been performed appropriately and rigorously? 

Reviewer #1: Yes

Reviewer #2: No

3. Have the authors made all data underlying the findings in their manuscript fully available?

Reviewer #1: No

Reviewer #2: Yes

4. Is the manuscript presented in an intelligible fashion and written in standard English?

Reviewer #1: Yes

Reviewer #2: No

5. Review Comments to the Author

Reviewer #1: Rodriguez-Enrigquez et al report a molecular modeling study of the M protein of PEDV. The homology model was build, the potential epitopes were predicted and the binding energy between the putative epitopes and the swine MHC-I were calculated. Because the experimental structure is not available yet, this study provides valuable structural information on this important protein. I would recommend this study if the authors address the following minor points:

1). Is there the M-gene knockout study to decipher the precise biological role of the M protein? Any literature evidence to support that the M protein is a good therapeutic target?

2). The M protein structure model should be placed in the membrane. The author can try OPM server (https://opm.phar.umich.edu/).

Reviewer #2: The manuscript reports a study aimed at predicting the 3D structure of the PEDV M protein. Then, the model has been utilized to derive its immunogenic profile using in-silico approaches.

In principle, the manuscript is an interesting demonstration of the usefulness of the application of in-silico methods to characterize the structure and function of a protein. However, in my opinion, there are several flaws that need to be fixed.

In general, the manuscript is difficult to read and should be reorganized and shortened. Only relevant information should be reported. For example, several paragraphs in the Materials and methods can be merged.

Major points:

line 93: authors claim that three groups have been identified in the tree. This is not clear from Fig. 1 . Please, explain.

Line 107. ambiguous positions: not very clear. Please, explain

Lines 112-126: all these data appear not strictly necessary. The paraghraph may be shortened

line 128: Please add a reference for the CAMEO project

line 138: it is not clear how the new PEDV sequence has been selected for modelling

line 149: The Table is not essential.

line 188. Fig. 2 legend is split. Is that correct?

line 194. Table 2 can be moved to the Supplementary data

line 287: Prediction of T-cell epitopes for the PEDV M protein: these methods are intrinsically inaccurate. Indeed, different prediction are obtained for different models of the same protein (Fig. 4), It may be interesting to show the consensus prediction.

Also, peptide docking is problematic. Indeed, Autodock Vina has not been specifically conceived for protein-peptide docking. Authors should explain better how they assigned binding energies (Table 3). A standard deviation should be calculated for each energy to account for the stocastic nature of the Vina algorithm. As a reference test, they should also calculate the binding energy of peptides not predicted to bind the SLA-01:0401 allele.

Minor points:

Please, reformat carefully all references (for example, line 105)

Figure 3 should be moved to Supplementary data.

6. PLOS authors have the option to publish the peer review history of their article (what does this mean?). If published, this will include your full peer review and any attached files.

Reviewer #1: **Yes: **Chun Wu

Reviewer #2: No

---

## [Author Response · Author response to Decision Letter 0]

19 Nov 2021

Dear Reviewers and Academic editor

I hope you are well. The authors apply all major and minor suggestions to the manuscript indicated by the reviewers and the academic editor. Many changes were made, so they are not attached in this communication window due to the number of points reviewed and corrected, but all comments and suggestions are answered in the Response to Reviewers file.

The article was shortened to include only the most relevant information as suggested by reviewer 2. The 3D M protein model was analyzed in the context of the membrane using the OPM/PPM server as suggested by reviewer 1. We respond to the comment related to the revision of the prediction of T cell epitopes as well as the proposed molecular docking. Only 5 figures and 2 tables were selected to be included directly in the revised manuscript and 4 complementary information files were selected to be presented in this new version of the manuscript. The references were reviewed and corrected. Figures were adjusted at a resolution of 300 dpi. 

We deliver the restructured work for evaluation and possible publication.

I really appreciate your kind attention.

Corresponding author

---

## [Editor Report · Decision Letter 1]

1 Dec 2021

PONE-D-21-16531R1Predicted 3D model of the M protein of Porcine Epidemic Diarrhea Virus and analysis of its immunogenic potentialPLOS ONE

Dear Dr. Rosas-Murrieta,

Thank you for submitting your manuscript to PLOS ONE. After careful consideration, we feel that it has merit but does not fully meet PLOS ONE’s publication criteria as it currently stands. Therefore, we invite you to submit a revised version of the manuscript that addresses the points raised during the review process.

Your revised manuscript was modified and addressed the concerns of the reviewers, as wells as my previous comments on the original manuscript. Since PLOS One does not perform final copy-editing of accepted manuscripts, I request that all changes must be done prior to acceptance. Upon reading your manuscript, I came upon a number of typographic errors that must be corrected. The list below is likely not complete and I recommend that you carefully check the complete manuscript for grammar and typographic errors. I would strongly recommend that the manuscript should corrected by a native English speaker, to make the text more fluent, thereby increasing the impact of your data on the reader.

Some specific editorial comments to address:

Title: update your title on manuscript title page. Currently it is duplicated with a second version, which seems to be the short title. Make sure both titles are separate to avoid confusion.L40 : remove “obtained”

L55: replace “the virus” by “viral”

L64: add pace before “and”

L114: superposition should be superimposition

L166:  “that we have genetically characterized in its entirety by our research group.” should be either “that we have genetically characterized in its entirety.” or “that was genetically characterized in its entirety by our research group”

L173: replace “being” by “showing” or by “being the most similar”

L 174: M proteins

L185; The sentence is unclear. Change to:

“Similarly, a value of depth/hydrophobic thickness of 20.8 ±1.9 Å, with a ΔGtransfer of -18.8 kcal/mol and tilt angle de 25±0 ° was calculated for the 3D 2013MMV M protein model.”

L195: the figure legend for panel 2D mentions blue pseudo atoms when they are red in the figure.  Also confirm that the color description of pseudoatoms in figure 2E is correct. Also checks lines 365-367.

L205-206: avoid using square brackets [0.1] for values since they can be mistaken for references. “must be in a range between 0 and 1”

L236: replace “were compared finding” with “displayed”

S2 Table: there are no double ** in the table, but it is mentioned as a footnote.

L247: superimposition

L275: servers

L287 and 296: Feig lab

L291: epitopes

Table S3: last part of the table, epitope number 3: the word model is inserted in the sequence (near V70), it probably does not belong there. If it does, please explain.

Table S4:  adjust width of fourth column so that “propensity” is on the same line. Otherwise, use proper hyphenation.

L328: structure of the M protein of PEDV is unknown

L337: represent the M protein

L339: the analysis included the genogroups

L353: remove space before “The”

L354: change Fig2 to Figure 2

L381: “can interact” instead of “interacts” since direct interactions with each cellular proteins were not directly demonstrated.

L395: period after [39]

L397: allow making

L398: allow the design of drugs

L401: replace “ the PEDV virus” with “PEDV”

We look forward to receiving your revised manuscript.

Kind regards,

Claude Krummenacher, PhD

Academic Editor

PLOS ONE
---

## [Author Response · Author response to Decision Letter 1]

16 Jan 2022

Dear Reviewers

I hope you are well. The authors applied to the manuscript all the suggestions indicated by the academic editor. Typographic errors have been corrected, but are not attached to this communication window due to the number of points corrected. All comments and suggestions are answered in the Response to Reviewers file. The English language has been checked and corrected by an English speaker. 5 figures and 2 tables, and 4 complementary information files were included in this new version of the manuscript. Figures were adjusted at a resolution of 300 dpi. We deliver the restructured work for evaluation and possible publication.

I really appreciate your kind attention.

Corresponding author

---

## [Editor Report · Decision Letter 2]

24 Jan 2022

Predicted 3D model of the M protein of Porcine Epidemic Diarrhea Virus and analysis of its immunogenic potential

PONE-D-21-16531R2

Dear Dr. Rosas-Murrieta,

We’re pleased to inform you that your manuscript has been judged scientifically suitable for publication and will be formally accepted for publication once it meets all outstanding technical requirements.

Kind regards,

Claude Krummenacher, PhD

Academic Editor

PLOS ONE
---

## [Editor Report · Acceptance letter]

31 Jan 2022

PONE-D-21-16531R2 

Predicted 3D model of the M protein of Porcine Epidemic Diarrhea Virus and analysis of its immunogenic potential 

Dear Dr. Rosas-Murrieta:

I'm pleased to inform you that your manuscript has been deemed suitable for publication in PLOS ONE. Congratulations! Your manuscript is now with our production department. 

Kind regards, 

on behalf of

Dr. Claude Krummenacher 

Academic Editor

PLOS ONE